# Why Bile Acids Are So Important in Non-Alcoholic Fatty Liver Disease (NAFLD) Progression

**DOI:** 10.3390/cells8111358

**Published:** 2019-10-30

**Authors:** Aline Gottlieb, Ali Canbay

**Affiliations:** 1Department of Physiology, Johns Hopkins University, Baltimore, MD 21224, USA; aline.gottlieb@jhmi.edu; 2Department of Medicine, Ruhr University Bochum, University Hospital Knappschaftskrankenhaus Bochum, 44892 Bochum, Germany

**Keywords:** bile acids, bile acid metabolism, NAFLD/NASH therapy, lipid metabolism, gut-liver axis, UDCA, OCA

## Abstract

Non-alcoholic fatty liver disease (NAFLD) is a complex disease, affecting not just the liver, but also all other organs in the body. Despite an increasing amount of people worldwide developing NAFLD and having it progress to non-alcoholic steatohepatitis (NASH) and potentially cirrhosis, there is still no approved therapy. Therefore, huge efforts are being made to find and develop a successful treatment. One of the special interests is understanding the liver–gut axis and especially the role of bile acids in the progression of NAFLD. Farnesoid X receptor (FXR)-agonists have been approved und used in other liver diseases, such as primary biliary cholangitis (PBC) and primary sclerosing cholangitis (PSC), and have shown signs of being able to decrease inflammation and potentially steatosis. This review will mainly focus on targets/ligands that play an important role in bile acid metabolism and give an overview of ongoing clinical as well as pre-clinical trials. With the complexity of the issue, we did not aim at giving a complete review, rather highlighting important targets and potential treatments that could be approved for NAFLD/NASH treatment within the next few years.

## 1. Introduction

### 1.1. Non-Alcoholic Fatty Liver Disease (NAFLD) Pathogenesis and Significance

Non-alcoholic fatty liver disease (NAFLD) represents the most common cause of chronic liver disease in developed countries worldwide [1,2]. NAFLD is considered to be the hepatic manifestation of the metabolic syndrome [3]. It can exacerbate from a fatty liver to non-alcoholic steatohepatitis (NASH). 

NASH is defined by macro vesicular fat accumulation and additionally by hepatic ballooning and lobular inflammation [4]. Approximately a fifth or a sixth of NASH patients will develop a cirrhosis [5]. NAFLD/NASH is supposed to become the leading cause of liver transplantation by 2020 [6]. Unfortunately, the pathogenesis of NAFLD/NASH development and progression remains only partially understood. Being a complex disease, interacting with other organs in the body, more attention has been focused on the interaction, for instance, of the intestine–liver–adipose tissue axis for its development. Several different pathways have been proposed over the past years on how NAFLD progresses, however it seems rather likely that NAFLD development depends on multiple hepatic insults via several different pathways [7]. One of them is that an altered lipid metabolism eventually leads to an increased fat accumulation by hepatocytes, causing oxidative stress and with that, cellular damage. Another way could be through adipocytes which release cytokines inducing inflammation and fibrosis within the liver. Bile acids (BA) for example have been discovered to play a pivotal role when it comes to insulin sensitivity as well as metabolic homeostasis [8,9]. In recent studies, a correlation of BA levels with NASH severity in obese individuals as well as a free fatty acid induced dysregulation of BA signaling in NASH, has been described [10]. Alterations in fecal and serum BA levels (represented as an increase in taurine- and glycine-conjugated BAs as well as an increased amount of secondary BAs) in NASH were confirmed later on [11,12]. Secondary BAs can be harmful and hence, an increase of secondary BAs can cause, partially, repeated insults of inflammation that could ultimately contribute to the progression of NASH. 

Even though NAFLD itself is asymptomatic and only sometimes affects liver function, it can progress to end-stage liver disease and as of right now, there are no approved therapies for NASH.

### 1.2. Bile Acids—from Synthesis over Transport to Recovery

BAs are synthesized from cholesterol in the liver and are a major component of bile. 

There are two ways BAs can be synthesized: First, there is a classical pathway (also called neutral pathway), and 75% of BAs are synthesized this way. The first step, which is the rate- limiting step, is catalyzed by the enzyme cholesterol 7α-hydroxylase (CYP7A1) to produce 7 α--hydroxycholesterols. This enzyme is solely expressed in the liver. Cholic acid (CA) is an example of a BA that is produced here [13].

The first step of the alternative pathway is initiated by the enzyme sterol 27-hydroxylase (CYP27A1). It is a mitochondrial enzyme that is more widely expressed and found also in macrophages and various tissues. 25-Hydroxycholesterol and 27-hydroxycholesterol are intermediate products of this alternative pathway. Chenodeoxycholic acid (CDCA) for example is a synthesized product of this pathway. There are publications indicating a shift towards the alternative pathway in NAFLD [14,15].

CA and CDCA are called primary BAs. They are conjugated with either glycine or taurine, even though glycine is more common [16].

Once these BAs are produced, they are being combined with other substances, such as cholesterol, phospholipids, and water and they are stored as bile in the gallbladder. After food consumption, they are released into the duodenum in response to cholecystokinin (CKK). BAs are necessary to emulsify lipids and absorb fats and fat-soluble vitamins. Most of them (95%) are recirculated to the liver from the terminal ileum. Only a small part enters the colon where they are either dehydroxylated or deconjugated by bacteria to form the so-called secondary BAs: deoxycholic acid (DCA) or lithocholic acid (LCA). LCA is insoluble and excreted whereas DCA is reabsorbed and recycled. This enterohepatic circulation involves several transporters, such as the apical-sodium dependent bile acid transporter (ASBT) for example [17]. There are even a few BAs that are taken back into circulation but cannot be recycled. These BAs will be excreted in urine [18].

### 1.3. Relationship of NAFLD and Bile Acids

There is no real epidemiological evidence as of right now, directly linking the development of cholestasis solely based on NAFLD/ NASH [19]. Usually, multifactorial components will cause cholestasis. Despite the lack of epidemiological evidence, it is discussed whether BAs could represent one of the so often debated about “second hits” in NAFLD progression. It has been shown in NAFLD/NASH patients that moderate elevations of serum levels of gamma glutamyl transferase (GGT), alkaline phosphatase (AP), and total bile acid (TBA) are common [10,20]. Furthermore, it has been described that the BA composition is changed in NAFLD/NASH patients.

Since that discovery, BAs have been studied intensively as a therapeutic target for NAFLD prevention and/or treatment. There is evidence, that NAFLD/NASH might cause a potential shift in pathways of patients when liver samples were compared to healthy livers: mRNA levels of CYP7B1 levels were increased, and CYP8B1 levels were decreased [15].

Metabolomic analyses have revealed significantly increased serum levels of glychochendeoxycholate, glycholate, and taurocholate serum levels in patients with NASH compared to healthy controls [21].

A transcriptome analysis, on the other hand, comparing the BA profiles of NASH patients and healthy patients only showed a correlation with the metabolic phenotype, especially the insulin resistance (IR), with no correlation to liver necroinflammation [22].

Part of the ongoing research is to describe changes of BA patterns. In a study in twins, it has been found that total serum BAs do not differ significantly between NAFLD vs. non-NAFLD and NAFLD vs. NASH, but they were significantly perturbed progressively as liver fibrosis increased [23]. 

BAs may be protective against NAFLD progression through the activation of the farnesoid X receptor (FXR). It is a theory to say, that with an increasing destruction of liver tissue, the amount of abnormal BA synthesis as well as secretion is increased [24]. A reduced flow will lead to accumulation and the changed constitution of that bile could thereby cause liver injury [25]. 

Changes of hepatic as well as ileal BA transporter expression influences NAFLD. The bile-salt export pump (BSEP) is the primary transporter of bile acids from the hepatocyte to the biliary system [26]. It is in the hepatocyte canalicular membrane and represents the rate limiting step in the secretion of bile salts by the liver. BSEP is critical for bile salt dependent bile flow and a normal enterohepatic circulation of bile salts from the distal intestine back to the liver. Mutations or defects of BSEP lead to the development of cholestasis [26]. In studies of BSEP in NAFLD, it has been shown that a reduced expression of bile-salt export pump (BSEP) was significantly correlated with the degree of NAFLD [27], and overexpression of BSEP lead to hepatic lipid accumulation [28]. 

## 2. Typical Ligands for Bile Acids

BA targeted therapy is based on their receptors and ligands. The regulation of BAs on other pathways occurs mainly through the activation of nuclear hormone receptors (NHRs) such as the farnesoid X receptor (FXR), pregnane X receptor (PXR), and vitamin D receptor (VDR) [29,30]. For the activation of these receptors, it is necessary that they interact with the retinoid X receptor as a heterodimer (RXR) [31]. Afterwards, they act on various regulatory regions by binding to hormone response elements causing several genes to be up or downregulated in their transcription. Most BAs and the secondary BA, and LCA, are binding to FXR and PXR respectively [32].

### 2.1. FXR

FXR plays an important role in BA, glucose, and lipid metabolism. It is most highly expressed in the liver, ileum, kidneys, and adrenal glands. The strongest activator is CDCA, followed by DCA, CA, and LCA [33]. Despite the fact that LCA is a weak activator of FXR, it is a very strong downregulator of its function and thus could be termed a partial agonist [24]. 

Controlling the synthesis and enterohepatic circulation of BAs, represents the major function of FXR. In the liver, bile flow is prompted through FXR activation [34]. After being activated through BA in the ileum, the expression of fibroblast growth factor 15/19 (FGF15/19) increases, leading to an activation of fibroblast growth factor receptor 4 (FGFR4) in hepatocytes (Figure 1). This is of great importance as FGF15/19 is an ileum-derived enterokine that governs BA homeostasis, regulates hepatic glucose metabolism, and stimulates protein synthesis. If administered pharmacologically or expressed transgenically in mice, FGF19 increases hepatic lipid oxidation, reduces lipogenesis and protects from hepatosteatosis. A lack of it causes impaired liver regeneration. A lack of FGF 15 results in increased hepatic steatosis and in the development of endoplasmic reticulum (ER) stress in the liver of mice fed a high fat diet [35].

The activation of FGFR4 together with the activation of small heterodimer partner (SHP) in the liver, inhibits CYP7A1. In order to downregulate *CYP7A1,* intestinal-specific FXR is required. On the other hand, the liver-specific FXR plays an important role in the repression of the expression of *CYP8B1* [36]. 

One of the mechanisms for BAs to suppress their own synthesis is through FXR [31]. 

FXR influences the hepatic lipid homeostasis via SHP, which reduces the expression of SREBP1 [37]. It also regulates the expression of peroxisome proliferator-activated receptor PPAR-α, a regulator of triglyceride metabolism, that can induce free fatty acid β-oxidation [38]. 

### 2.2. Takeda G-Protein-Coupled Receptor 5 (TGR5)

TGR5 is a G-protein coupled membrane receptor that BAs activate. LCA is considered to be the strongest activator of TGR5 among the main primary and secondary BAs [39]. It is widely expressed in the body, for example in the gallbladder, ileum, colon, liver (not in hepatocytes though [40]), brown adipose tissue (BAT), nervous system, and muscle [39]. Besides being involved in BA, glucose, and lipid metabolism, interestingly, it also plays a role in increasing the induction of *BAT* expression in thermogenesis and energy release via browning of white adipose tissue and [41]. TGR5 knockout leads to a changed BA composition [42] and has anti-inflammatory properties by for example by inhibiting nuclear factor kappa-light-chain-enhancer of activated B cells (NF-κB) [43] or LPS-induced production of inflammatory cytokines in macrophages [44]. In another experiment, TGR5 knockout mice received a sleeve gastrectomy and it was shown how TGR5 is important in order to decrease hepatic steatosis, improve glucose control, and increase the energy expenditure post-surgery [45]. 

## 3. Effects from Bile Acids on Different Metabolic Functions in the Body

### 3.1. Clinical Manifestation of Dysregulated BAs in NAFLD

The risk of hepatic injury is increased by dysregulated BA metabolism in adult NAFLD patients [11]. It was also shown that NASH patients have higher post-prandial release of BAs, and thus are more susceptible to damage caused by secondary BAs (resulting from bacterial actions in the colon) [12].

As previously mentioned, cholestasis is a complex disease. Typical symptoms are jaundice, itching, urine discoloration, and fecal whitening. Usually bilirubin, total cholesterol (TC), total triglycerides (TG), and GGT are elevated. However, serum alanine aminotransferase (ALT), aspartate aminotransferase (AST), and alkaline phosphatase (ALP) can also be increased. 

### 3.2. Bile Acid Effects on Glucose Metabolism

BAs help regulate the glucose metabolism via FXR and TGR5, but the exact mechanism is still unclear, and studies show differing results. 

FXR, for instance, decreases glycolysis and hepatic gluconeogenesis, and increases glycogen synthesis [46]. FXR modulates the expression of phophoenolpyruvate carboxykinase (PEPCK) and glucose-6-phosphatase (G6P), and with that, influences the glucose metabolism [47]. Contrary results have been published though, where BAs cause an increase of insulin resistance [48].

The activation of TGR5 through BAs results in increased secretion of GLP-1 and decreased insulin resistance [49].

Insulin sensitivity is also partially regulated by BA. For that, BAs need to communicate with white adipose tissue (WAT). There is a group called “adipokines” (adipose tissue cytokines) which are secreted by adipocytes and function as cell signaling proteins [50]. Since the first discovery, hundreds of different adipokines have been discovered [51]. The major representatives are adiponectin, leptin, and resistin) and they all seem to play a role in liver injury [52] Adiponectin is produced in adipocytes and has anti-inflammatory and anti-fibrotic qualities. It induces uptake of glucose in several tissues, which in return decreases gluconeogenesis in the liver and inhibits production of pro-inflammatory cytokines like interleukine 6 (Il-6) [53]. Adiponectin seems to be negatively regulated in BAs, as patients with NASH have high levels of BAs but low levels of adiponectin [10]. Later on in the disease, it is suspected that adiponectin increases again, as a potential marker for NASH progression toward cirrhosis in humans [54]. The specific interaction between BAs and adipokines needs to be further elucidated [55]. 

Leptin has multiple metabolic functions, for example inhibition of food intake, stimulation of fatty acid oxidation in the liver and skeletal muscle, stimulation of glucose uptake in skeletal muscle, stimulation of insulin secretion, stimulation of proinflammatory cytokines, suppression of fatty acid biosynthesis, and suppression of hepatic glucose production [52,56]. 

Resistin could be a link between insulin resistance and obesity; its release reduces peripheral insulin sensitivity, increases endogenous glucose production by the liver, induces insulin resistance, and stimulates proinflammatory cytokines (e.g., IL-6 and tumor necrosis factor (TNF)-α) [52].

A study conducted by Huang et al., examining 18 obese patients with type 2 diabetes mellitus, who underwent sleeve gastrectomy (SG), tried to elucidate the effects of SG in treatment of NAFLD patients. A year after SG, the total BA level and fatty liver index were significantly decreased in NALFD improvers and T2DM complete remitters, while FGF19 levels were increased. This could suggest, that FGF19 and total BAs might play a role in T2DM remission and NAFLD improvement [57].

### 3.3. Role of Bile in Lipid Metabolism

Triglyceride metabolism is regulated via the FXR/SHP-signaling pathway through Bas, thus, the synthesis of triglycerides in the liver is regulated by a feedback mechanism. Many enzymes are involved in lipogenesis, for example acetyl-CoA carboxylase (ACC), fatty acid synthase (FAS), and glucose- 6-phosphatase (G6Pase). Their function is regulated by SHP. The most important regulator of fatty acid and triglyceride biosynthesis is the sterol binding element binding protein 1 (SREBP1). SREBP1 is inhibited by SHP [37]. Via activation of sphingosine-1 phosphate receptor 2 (S1PR2), BAs also regulate lipid metabolism In a knockout mice model fed with a high fat diet (HFD), it was shown that hepatic lipid accumulation was increased [58].

Taurine-conjugated CA (TCA) activates several pathways, such as the S1RP2-ERK1/2 pathway and protein kinase B (AKT) pathways, which are involved in glucose and lipid metabolism pathways [58,59].

CDCA-treatment of human hepatocytes showed changes in expression of genes that regulate lipid homeostasis, such as APOL3, FABP3, LDLR and SLC27A2 in a CDCA-dependent manner [60]. Similarly, CDCA in addition to DCA and LCA increases LDL receptor gene expression via the mitogen-activated protein (MAP) -Kinase pathway [61]. Several microRNAs, involved in lipid metabolism, were changed in their expression which could partially account for the effects of CDCA [60].

### 3.4. Role of Bile in Cholesterol Metabolism

Cholesterol is the substrate of BA synthesis and thus, it is metabolized when turned into BAs but as a result, it also accumulates when BA synthesis is inhibited. In order to have cholesterol secreted into the blood, BAs and phospholipids are required as well [62].

Cholesterol can either be taken up into the liver by low-density lipoprotein (LDL) or synthesized de novo from acetate [13].

Another way for cholesterol to enter the liver is the so-called reverse cholesterol transport via high-density lipoprotein (HDL). The transporter responsible for the first step is ATP Binding Cassette Subfamily A Member 1 (ABCA1) which transports cholesterol from peripheral tissues to apolipoproteins [63]. The uptake of cholesterol ester from HDL into the liver is regulated by scavenger receptor class B type I (SR-B1) [64]. Cholesterol levels can additionally be reduced by FXR which induces SR-B1 expression to enhance HDL removal from the blood into the liver [65].

### 3.5. Role of Bile in the Intestine, and with Microbiota 

Hepatic FXR is more important in the prevention of hepatic lipid accumulation than intestinal FXR [66].

Interestingly, intestinal bacteria seem to influence fat accumulation in hepatocytes which is independent of body fat. BAs are very important regulators of intestinal bacteria balance. The number of gram positive bacteria seems to be reduced by low BA levels [67]. A changed composition of BAs could be an important regulator of intestinal bacteria. Secondary BAs that are influenced through intestinal bacteria could activate FXR and TGR5 with a stronger extent than primary BAs [68]. 

It has been shown, that intestinal bacteria differ significantly between healthy individuals and NAFLD patients. That being said, it appears plausible that intestinal bacteria could play a role in the regulation of immune function during progression of NAFLD. Furthermore, the changes of intestinal bacteria could have an effect on the development of fibrosis, hepatic steatosis, and inflammation [69]. This indicates that an upkeep of intestinal microbiota via BA metabolism should prevent NAFLD. Evidence accumulates that primary and secondary BAs are increased in NAFLD patients. The FXR antagonist DCA was increased in these patients, while the FXR agonist CDCA was decreased. That can, in part, explain the suppression of FXR-mediated and FGFR4-mediated signaling. Additionally, taurine and glycine metabolizing bacteria were increased in the gut of NAFLD patients, which reflects an increased secondary BA production [46]. 

In an in vivo study of mice being fed a high fat diet (HFD) and being treated with green tea polyphenol (epigallocatechin-3-gallate, EGCG), a significant inhibition of weight gain was observed in those mice treated with EGCG compared to the control. Also, fatty lesions and triglyceride content was decreased in these mice livers. Interestingly, the authors were able to show a significant change of gut microbiota towards an increase of *Adlercreutzia, Akkermansia* and *Allobaculum* and a significant decrease in abundance of *Desulfovibrionaeceae* [70].

A prospective study of 127 NAFLD patients examined the associations of FGF19, C4, and BA diarrhea. An increased hepatic BA production and diarrhea were associated with an increased NAFLD score, but not a low FGF19, indicating dysregulation of the FXR-FGF19 axis. Metformin was an important factor in a subgroup lowering FGF19, and resulting in BA diarrhea [71]. 

## 4. Pharmacotherapies

### 4.1. FXR Agonists

Most of the drugs have been tested in the field of FXR agonists. The likely most important representative in that field is obeticholic acid (OCA), a semi-synthetic derivative of CD which will be discussed further now. 

### 4.2. OCA

OCA (INT-747) is used in many liver diseases such as biliary atresia, primary biliary cholangitis (PBC), NAFLD/NASH, and primary sclerosing cholangitis (PSC). OCA treatment significantly reduces hepatic gluconeogenesis and lipogenesis. In addition, the hepatic inflammation can be inhibited and, through OCA induced FXR activation, the intestinal inflammation can be inhibited. 

After having competed three promising phase II studies, OCA is now being tested in two different phase III trials. One of them is called the “REGENERATE” trial (NCT02548351), a multi- center, double-blind trial evaluating the safety and efficacy of OCA in 2380 NASH patients over 18 months. One of the biggest concerns of OCA is the worsening of the lipid profile, as noticed in phase II studies [72]. 

Primary endpoints include the amount of OCA treated patients relative to placebo achieving at least one stage of liver fibrosis improvement with no worsening of NASH, or the proportion of OCA treated patients relative to placebo achieving NASH resolution with no worsening of liver fibrosis.

The other phase III trial is called “REVERSE” (NCT03439254). It is testing the efficacy of OCA in 900 NASH patients over 18 months. The primary outcome is to look at an improvement in fibrosis by at least one stage with no worsening of NASH. Secondary outcomes are to look at the percentage of subjects with improvement in fibrosis by at least two stages, and the percentage of subjects with NASH resolution.

There is also a small study being conducted this year (NCT03836937) examining the role of OCA in NAFLD patients with elevated ALT levels. 

### 4.3. Ursodeoxycholic Acid (UDCA)

UDCA is a selective FXR agonist approved for therapy of PBC and PSC. In these diseases it shows anti-cholestatic and anti-apoptotic effects. 

In a study with 40 morbidly obese patients that either received 20 mg UDCA/day (d) or no treatment three weeks prior to their bariatric surgery, UDCA treatment stimulated BA synthesis by reducing FGF19 and FXR activation which resulted in cholesterol 7α hydroxylase induction. The enhanced BA formation depleted hepatic and LDL cholesterol [73].

However, in several phase II studies, it failed to improve the histology in NAFLD or NASH patients, when administered as the only treatment [74]. 

A potential option for it could be in combination with other targets. A study combining UDCA with losartan, an angiotensin receptor blocker, decreased the NAFLD activity score (NAS) and seemed to have a synergistic effect in decreasing hepatic fibrosis in rats [75].

Another study from Iran, conducted in rats, showed a synergistic effect of curcumin and UDCA in the treatment of NAFLD [76]. This paper focused more on the inflammation and necrosis/apoptosis axis. 

Table 1 summarizes ongoing studies of FXR and PPAR α-agonists. 

### 4.4. PPARα Agonists

PPARα ligands (such as fenofibrate and bezafibrate) inhibit triglyceride synthesis efficiently [77]. Fibrates reduce the level of ALP in patients with abnormal lipid metabolism [78]. In a study comparing ciprofibrate treatment in PPARα null mice to wildtype mice, CYP7A1 and CYP27A1 were only decreased in wildtype mice, indicating that the BA biosynthesis could be inhibited by activation of PPARα by fibrin [79]. Furthermore, PPARα ligands show anti-inflammatory activity. They seem to repress interferon gamma (IFN-γ) and IL-17 production in cluster of differentiation 4(CD4+) T- cells, increase the secretion of chemokines in epithelial cells and reduce the expression of cell adhesion molecules in endothelial cells [80]. In NAFLD patients, one of the major goals is to reduce the accumulation of triglycerides in hepatocytes. Fibrates are able to upregulate enzymes related to FA oxidation, for example acyl-CoA synthetase [81]. All PPAR ligands are involved in either β-oxidation, lipid, and/or glucose metabolism as shown in Figure 2. 

At the time this article was written (September 2019) two phase II studies were active and recruiting to study the effects of PPARα in NAFLD patients. 

One of them is the open label “EVIDENCE III” (NCT03639623) trial. It is testing Saroglitazar, a dual PPARα and –γ agonist, in liver transplant recipients. The primary outcome is to document the adverse effects. Secondary outcomes are looking for examples in the changes in hepatic fat, and changes in liver enzymes, glucose, and fructose levels. 

The other trial is testing IVA337 (Lanifibranor), a pan-PPAR agonist, in NASH patients (NCT03008070). Patients either receive 800 mg, 1200 mg or a placebo for 24 weeks. The primary outcome is the improvement of the SAF (steatosis, activity and fibrosis score). A responder will be considered by an improvement of at least 2 points in the score. 

A phase IV trial with Lobeglitazone (PPAR –γ agonist) was finished in 2016 and published in 2017 (NCT02285205), measuring the reduction of hepatic fat content via the controlled attenuation parameter (CAP). Despite these promising results, no further studies have been conducted or published since then. 

### 4.5. TGR5

TGR5, also known as G protein-coupled bile acid receptor 1 is a cell-surface receptor which is widely expressed in human tissue, including the intestine [82]. TGR5 agonists induce systemic release of glucagon-like peptides (GLPs) 1 and 2 and peptide YY (PYY) in intestinal L cells, a type of enteroendocrine cell abundant in the ileum and colon. [83]. GLP-1, an incretin, has insulinotropic effects in the pancreas to regulate glucose homeostasis, as well as extra pancreatic indirect metabolic effects [84]. The usage of TGR5 agonists have been hindered in recent years because of systemic side effects, such as gallbladder emptying. It was recently shown that the development of a more selective TGR5 agonist is possible. One of them is called RDX8940, which showed improved liver steatosis and insulin sensitivity in a mouse model of NAFLD [85] without systemic side effects. 

There is also a FXR/TGR5 dual agonist, INT-767, which significantly counteracts high-fat diet -induced liver and fat alterations in a mouse model, restoring insulin sensitivity and inducing pre-adipocyte differentiation toward a metabolically healthy phenotype [86].

At the time this article was composed, no human studies were registered at www.clinicaltrials.gov.

### 4.6. Bile Acid Conjugates

Studying the effects on BA administration has been a more recently re-discovered field. The idea behind it, is to increase the BA flow and hence support all their positive effects. 

Except for a study conducted in Israel in 2014, where patients received Aramchol and showed a significantly lower liver fat content and increased adiponectin levels [87], no other human study has been published as of now. 

A semi-synthetic BA derivative, called TC-100, showed promising in vivo results. It seemed to have a better binding ability to FXR compared to OCA and similar effects on the regulation of FXR target genes [88]. Since its first description however, nothing new has been published. 

### 4.7. Apical Sodium-Dependent Bile Salt Transporter (ASBT) Inhibitors

ASBTs are another example of targets that have been discovered more recently. ASBTs are the main transporters to promote reabsorption of BAs from the intestine into enterohepatic circulation. Inhibiting this process could increase the excretion of BAs, thus increasing BA synthesis and consequently cholesterol consumption [89].

SHP626 (volixibat) is an example of an ASBT, that was supposed to achieve FDA approval on a fast track and its phase II study was terminated in 2018. Interestingly, another in vivo study in mice was published in 2019, showing that volixibat significantly increased the total amount of BA in feces. Administered at the highest dose, volixibat significantly constricted the HFD-induced increase in hepatocyte hypertrophy, hepatic triglyceride and cholesteryl ester levels, and mesenteric white adipose tissue deposition. The NAS in volixibat-treated mice was significantly lower than in the HFD controls [90]. 

In 2018, a different study was conducted with IMB17-15. Hamsters were fed a HFD. It was shown that IMB17-15 suppressed FXR and FGF15/19 expression, which reduced Extracellular Signal-regulated Kinase (ERK) and c-Jun N-terminal kinase (JNK) levels. CYP7A1 activity was upregulated and the expression of PPARα was increased [91]. 

## 5. Summary and Outlook

Finding an effective NAFLD/NASH treatment has been and will be one of the biggest challenges in the field of hepatology. The reason for this is the complexity of the disease and the necessity to target more than just one problem in NAFLD. A useful combination needs to be found that can treat insulin resistance, changed lipid metabolism, inflammation, and the development of fibrosis. Stabilizing the bile acid metabolism to a more “normal” level that has been described in healthy controls seems to be an important part of the therapy. The research in that field is complicated by the complexity of the liver–bile–intestinal axis and the reduced overlap of intestinal bacteria between animals and humans. It seems as though scientists have found a few very promising targets, such as UDCA and OCA in order to tackle these. The next few years will show if these targets are as effective as hoped for, in the phase III trials. 

Another action that needs to be taken, is to understand better, which patients, out of a million affected with NAFLD, will be at risk to develop NASH, fibrosis, cirrhosis, and HCC, and might need a liver transplantation. For that we need successful prognostic non-invasive markers that will help to screen big cohorts of people in a fast and reasonable manner. 

## Figures and Tables

**Figure 1 cells-08-01358-f001:**
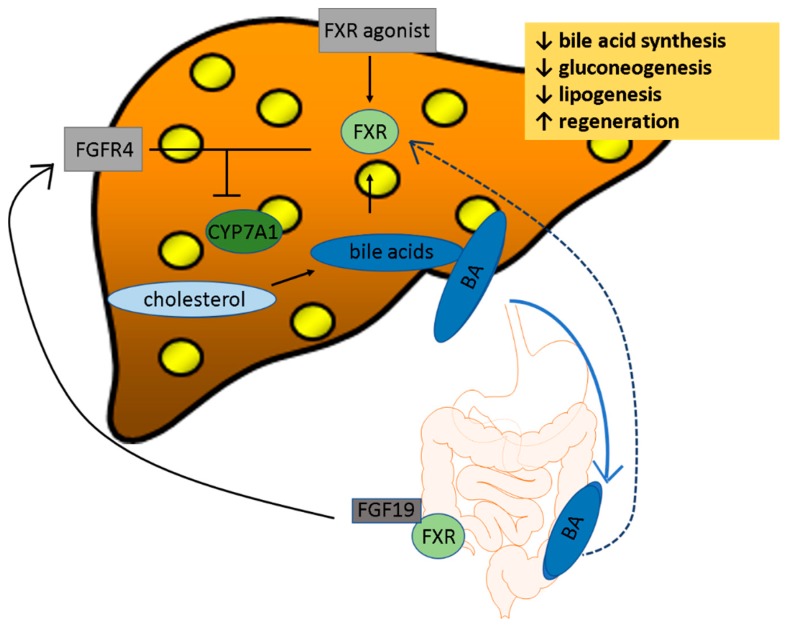
Illustration of how farnesoid X receptor (FXR) relation in bile acid homeostasis. FXR blocks bile acid (BA) synthesis through cholesterol 7α-hydroxylase (CYP7A1). Additionally, bile acid synthesis is blocked via the BA-dependent intestinal fibrolast growth factor (FGF19), which activates fibroblast growth factor receptor 4 (FGFR4) in the liver. There is also a negative feedback loop from BA on FXR.

**Figure 2 cells-08-01358-f002:**
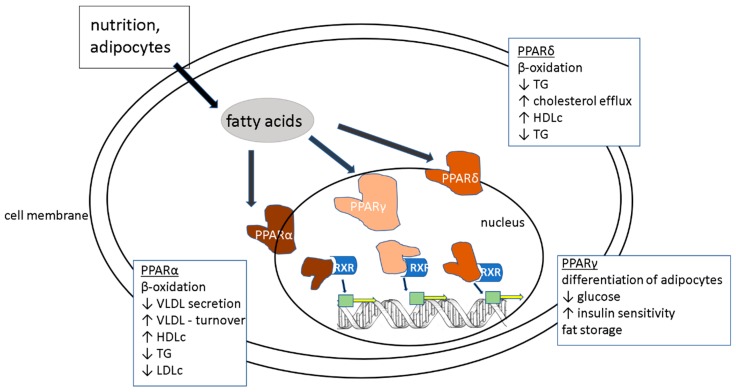
Simple illustration of the functions and effects of the different peroxisome proliferator-activated receptor (PPAR) subtypes. Through nutrition and from adipose tissue, fatty acids are being taking into cells. There they activate the different subtypes of PPARs, which then enter the nucleus and connect with retinoid X receptor (RXR) receptors. Binding of both leads to the inductions of transcription of several different genes involved and needed in lipid and cholesterol metabolism (not shown in detail). Abbreviations: High-, low, and very low- density lipoprotein (HDL, LDL, VLDL), Trigycerides (TG).

**Table 1 cells-08-01358-t001:** Selection of important ongoing studies of farnesoid receptor x (FXR) as well as peroxisome proliferator-activated receptor (PPAR) agonist. Non-alcoholic fatty liver disease (NAFLD), non-alcoholic steatohepatitis (NASH).

Drug Name/*Dosage*	Name of Study	Clinicaltrials.gov ID	Phase	Status
**FXR agonists**				
**OCA (Obeticholic Acid)**	The Farnesoid X Receptor (FXR) Ligand Obeticholic Acid in NASH Treatment Trial (FLINT)	NCT01265498	2	completed
	Study of INT-747 in Patients With Diabetes and Presumed NAFLD	NCT00501592	2	completed
	Combination Obeticholic Acid (OCA) and Statins for Monitoring of Lipids (CONTROL)	NCT02633956	2	completed
*10, 25mg*	Randomized Global Phase 3 Study to Evaluate the Impact on NASH With Fibrosis of Obeticholic Acid Treatment (REGENERATE)	NCT02548351	3	recruiting
*10mg, 10 or 25mg*	Study Evaluating the Efficacy and Safety of Obeticholic Acid in Subjects With Compensated Cirrhosis Due to Nonalcoholic Steatohepatitis	NCT03439254	3	recruiting
**PPAR agonists**				
**Saroglitazar** (PPARα and – γ agonist)	Safety, Tolerability and Efficacy of Saroglitazar Mg 4 mg in Liver Transplant Recipients With NAFLD	NCT03639623	2	recruiting
**IVA-337 (Lanifibranor)** (Pan-PPAR agonist)	Phase 2b Study in NASH to Assess IVA337 (NATIVE)	NCT03008070	2b	recruiting
**Lobeglitazone** (PPARγ agonist)	A 24 Week, Multicenter, Prospective, Open-labeled, Single-arm, Exploratory Phase 4 Clinical Trial to Evaluate the Safety and Efficacy of Lobeglitazone in Decreasing Intrahepatic Fat Contents in Type 2 Diabetes With NAFLD	NCT02285205	4	completed

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
