# Peer review of "Why Bile Acids Are So Important in Non-Alcoholic Fatty Liver Disease (NAFLD) Progression"

_cells, 2019, doi:10.3390/cells8111358_

Round 1

Reviewer 1 Report

Gottlieb's work is an ambitious revision work on a topic that promises to be of great interest for its clinical implications in a pathology with ever increasing socio-economic and health impacts such as NAFLD. However, as the authors themselves admit, currently, the level of evidence for the implications of bile acid metabolism in the pathogenesis and progression of NAFLD has not been highly supported. In any case, the work is certainly useful to indicate future research directions in this area.

However, some clarifications and further insights may improve the quality of the present work:

rows 98-100: the authors should better explain the role of BSEP and accurately verify the cited papers.

rows 114-119: the authors should investigate the role of FGF 15/19, due to its important implications in the gut / liver axis, metabolic regulation, insulin resistance and progression of fibrosis (see Avila MA and Berasain C).

rows 159-166: different adipokines are involved in the cross-talk between adipose tissue and liver. Perhaps this point deserves a somewhat more comprehensive treatment (see Carpino G and Gaudio E).

Author Response

Dear reviewer, 

thank you for your feedback to improve our manuscript. 

We hope to have succesfully edited our manuscript. 

Reviewer  1:
We thank the reviewer for his positive and diligent assessment of our manuscript as well as pointing out further aspects to improve the quality of it.

For information: The mentioned pages and rows apply to the manuscript version that is available online. Not the revised manuscript that has been uploaded after the letter to the reviewers.

„rows 98-100: the authors should better explain the role of BSEP and accurately verify the cited papers.“

The appropriate paragraph has now been edited to this (p 3, row 99-106):

“Bile-salt export pump (BSEP) is the primary transporter of bile acids from the hepatocyte to the biliary system [26]. It is in the hepatocyte canalicular membrane and represents the rate limiting step in the secretion of bile salts by the liver.  BSEP is critical for bile salt dependent bile flow and a normal enterohepatic circulation of bile salts from the distal intestine back to the liver. Mutations or defects of BSEP lead to the development of cholestasis [26]. In studies of BSEP in NAFLD, it has been shown that the reduced expression of bile-salt export pump (BSEP) was significantly correlated with the degree of NAFLD [27] and the overexpression of BSEP leads to hepatic lipid accumulation [28].”

rows 114-119: the authors should investigate the role of FGF 15/19, due to its important implications in the gut / liver axis, metabolic regulation, insulin resistance and progression of fibrosis (see Avila MA and Berasain C).

We tried to implement more information about FGF15/19 being an important factor for the gut/liver axis). The paragraph now reads like this (page 3, row 121-134):

“The major function of FXR is to control the synthesis and enterohepatic circulation of BAs. In the liver, bile flow is prompted through FXR activation [34]. For the downregulation of Cyp7a1, intestinal- specific FXR is required. After being activated through BA in the ileum, the expression of fibroblast growth factor 15/19 (FGF15/19) increases, leading to an activation of FGFR4 in hepatocytes (Fig.1),. This is of such importance, because FGF15/19 is an ileum-derived enterokine that governs BA homeostasis, regulates hepatic glucose metabolism, and stimulates protein synthesis. If administered pharmacologically or expressed transgenic in mice, FGF19 increases hepatic lipid oxidation, reduces lipogenesis and protects from hepatosteatosis. A lack of it causes impaired liver regeneration. A lack of FGF 15 results in increased hepatic steatosis and in the development of endoplasmic reticulum (ER) stress in the liver of mice fed a high fat diet [35].

The activation of FGFR4 together with the activation of small heterodimer partner (SHP) in the liver, inhibitsting CYP7A1.”

rows 159-166: different adipokines are involved in the cross-talk between adipose tissue and liver. Perhaps this point deserves a somewhat more comprehensive treatment (see Carpino G and Gaudio E).

We elucidated the different major adipokines further. This is the edited paragraph (page 5, 174-192):

“Insulin sensitivity is also partially regulated by BA. For that BAs need to communicate with white adipose tissue (WAT). There is a group called “adipokines” (adipose tissue cytokines) which are secreted by adipocytes and function as cell singalling proteins [50]. Since a first discovery, hundreds different adipokines have been discovered [51]. The major representatives are adiponectin, leptin, and resistin) and the all seem to play a role in liver injury [52] Adiponectin is an adipokine produced in adipocytes that has anti-inflammatory and anti-fibrotic properties. Adiponectin induces uptake of glucose in multiple tissues, which then decreases gluconeogenesis in the liver and inhibits production of pro-inflammatory cytokines like Il-6 [53]. Adiponectin seems to be negatively regulated in BAs, as patients with NASH have high levels of BAs but low levels of adiponectin [10]. Later on, in the disease, it is suspected that adiponectin increases again, as a potential marker for NASH progression toward cirrhosis in humans [54]. The specific interplay between BAs and adipokines needs to be further elucidated [55].

Leptin has multiple metabolic functions, for example inhibition of food intake, stimulation of fatty acid oxidation in the liver and skeletal muscle, stimulation of glucose uptake in skeletal muscle, stimulation of insulin secretion, and stimulation of proinflammatory cytokines, suppression of fatty acid biosynthesis, suppression of hepatic glucose production [52, 56].

Resistin could be a link between insulin resistance and obesity; its release reduces peripheral insulin sensitivity, increases endogenous glucose production by the liver, induces insulin resistance, and stimulates proinflammatory cytokines (e.g., IL-6 and TNF-α) [52].”"

Reviewer 2 Report

This review paper describes the roles of bile acids in regulation of different features of NAFLD and reviews recent clinical studies using BA in treatment of NAFLD.

Comments:

Several sections are confused and needs to be rewritten e.g: FXR, Role of bile in cholesterol metabolism, ...). In my opinion a clear separation of results reported in humans from animal models would improve the added value of this manuscript. Page 5, section "role of bile in cholesterol metabolism" is  describing everything from lipids to glucose. It is not clear which statements are relevant for cholesterol itself. Also cholesterol is hardly "degraded" to BA, I would use "metabolised" to BA Several abbreviations are not explained in the text (e.g. PBC, PBS, CA, ALP, GGT, HF, HFD, etc.). Some are not necessary as they appear only once  (e.g., NHR). Some are inconsistent: HF in HFD. Abbreviations of several genes are not explained and not consistent (e.g. CYP7a1, page 3, line 115). Inconsistency in writing thehydroxycholesterols: correct is 27-hydroxycholesterol, nevertheless, use one version throughout the manuscript. Overall manuscript must be improved by editing of English language. Page 3, section "FXR" it is very confusing, the role of hepatic FXR in BA regulation is therefore not clear. Page 2, section "Bile acids - from synthesis..." I do not agree that end products of alternative bile acids synthesis pathway are 27 - and 25-hydroxycholesterols. End products are bile acids (e.g.CDCA). Page 2, section "Bile acids - from synthesis...", statement: "There are several publications indicating a shift towards the alternative pathway in NAFLD" is supported by only one reference in hamster. Either change the statement or add references. Figure 1, SHP would also fit in the scheme. Page 7, section "PPARa agonists", ACO is not an official gene symbol for acyl-CoA synthetase, it is not clear which gene is refered. Page 8, title "Conjugates" refers to Bile acids conjugates? Page 9, line 342, NAS is probably NASH? Figure 2, A typo in the PPARdelta box: "cholesterin" efflux? Table 1 is of poor quality

Author Response

Dear reviewer, 

thank you for your comments and remarks to improve our manuscript. We hope you are satisfied with the adjusments we made to it.

We thank the reviewer for his honest assessment of our manuscript as well as pointing out further aspects to improve the quality of it.

For information: The mentioned pages and rows apply to the manuscript version that is available online. Not the revised manuscript that has been uploaded after the letter to the reviewers.

Several sections are confused and needs to be rewritten e.g: FXR, Role of bile in cholesterol metabolism, ...).

See further below to address these issues

In my opinion a clear separation of results reported in humans from animal models would improve the added value of this manuscript.

We fully understand the possible confusion that can be caused by mixing human and animal data within a paragraph. However, for this manuscript, we decided to present data according to the different treatment options and state very clearly each time, when we are talking about animal experiments. We hope that answer suffices the reviewer.

Page 5, section "role of bile in cholesterol metabolism" is describing everything from lipids to glucose. It is not clear which statements are relevant for cholesterol itself.

We adjusted this section to only cholesterol relevant parts by deleting the last two passages.

Also cholesterol is hardly "degraded" to BA, I would use "metabolised" to BA

We changed the word from degraded to „metabolized“

Several abbreviations are not explained in the text (e.g. PBC, PBS, CA, ALP, GGT, HF, HFD, etc.).

We kindly refer to the following pages in our manuscript:

PBC and PSC are first described on page 6 row 238 The authors are not sure what CA refers to. If it stands for cholic acid that abbreviation (as part of different cholic acids) is being explained several times, for instance page 2, row 59 and 69. To avoid confusion it has been added on the same page in row 54 when cholic acids are first mentioned. ALP and GGT are both first introduced on page 2 row 79. Some are inconsistent: HF in HFD HFD: High- Fat diet is first introduced on page 5 row 221. Out of 5 times the HFD is discussed, one time in our manuscript it said, “HF diet”. We corrected that mistake. Some are not necessary as they appear only once (e.g., NHR).

NHR is first mentioned and explained on page 3, row 103

Abbreviations of several genes are not explained and not consistent (e.g. CYP7a1, page 3, line 115).

CYP7A has been first explained on page 2 row 53 It has been brought to the attention correctly, that we had two different versions on how to write CYP genes in our manuscript. We made the necessary adjustments.

Inconsistency in writing the hydroxycholesterols: correct is 27-hydroxycholesterol, nevertheless, use one version throughout the manuscript.

We could not find any mistakes in the writing of hydroxycholesterols in our manuscript. If this is a pivotal point for the reviewer, we would kindly ask him/her to clarify what exactly he/she is pointing at and we will be happy to make the adjustments.

Overall manuscript must be improved by editing of English language.

Thank you for that remark. Our manuscript has been prove read by a native speaker. We hope to accomplish an improved manuscript by implementing your stylistic and content remarks.

Page 3, section "FXR" it is very confusing, the role of hepatic FXR in BA regulation is therefore not clear.

We re-arranged one of the paragraphs that could have been confusing in the section describing the role of FXR in BA regulation. We hope that makes this section more understandable.

„The major function of FXR is to control the synthesis and enterohepatic circulation of BAs. In the liver, bile flow is prompted through FXR activation [33]. For the downregulation of Cyp7a1, intestinal- specific FXR is required. After being activated through BA in the ileum, the expression of fibroblast growth factor 15/19 (FGF15/19) increases, leading to an activation of FGFR4 in hepatocytes (Fig.1), together with the activation of small heterodimer partner (SHP) in the liver, inhibiting CYP7Aa1. In order to downregulate CYP7A1, intestinal- specific FXR is required. On the other hand, Tthe liver- specific FXR plays an important role in the repression of the expression of CYP8Byp8b1 [34].“

Page 2, section "Bile acids - from synthesis..." I do not agree that end products of alternative bile acids synthesis pathway are 27 - and 25-hydroxycholesterols. End products are bile acids (e.g.CDCA).

In our previous version this sentence has been confusing. We corrected it to the following version: (page 2 Row 58-60)

“25- Hydroxycholesterol and 27- hydroxycholesterol are intermediate products of this alternative pathway. Chenodeoxycholic acid for example (CDCA) is synthesized product of this pathway.”

Page 2, section "Bile acids - from synthesis...", statement: "There are several publications indicating a shift towards the alternative pathway in NAFLD" is supported by only one reference in hamster. Either change the statement or add references.

We added one more source and took the strong word “several” out of the sentence.

„There are publications indicating a shift towards the alternative pathway in NAFLD [14, 15]. “

Figure 1, SHP would also fit in the scheme.

The reviewer is correct, that SHP would also fit into this schema. We only created this cartoon to explain the function of FXR- agonists. We hope that the reviewer is ok with us leaving SHP out in order to avoid confusion why we mention it there. We are also aware though, that we are leaving out certains facts and cannot mention everything that is related to that topic.

Page 7, section "PPARa agonists", ACO is not an official gene symbol for acyl-CoA synthetase, it is not clear which gene is referred.

Thank you for pointing out that mistake. We deleted the abbreviation in our manuscript.

Page 8, title "Conjugates" refers to Bile acids conjugates?

This is indeed too short and misleading, we changed it into Bile acid conjugates.

Page 9, line 342, NAS is probably NASH?

No, NAS stands for NAFLD- Activity Score. The abbreviation is first introduced on page 6, Row 267.

Figure 2, “A typo in the PPARdelta box: "cholesterin" efflux?”

We corrected the typo and changed it into cholesterol efflux."

“Table 1 is of poor quality “

Table 1 has been created with Power Point (standard table settings). We hope with the help of the graphics team, we reach a better resolution of the table in our manuscript.